# Clinical value of questionnaires & physical tests for patellofemoral pain: Validity, reliability and predictive capacity

**Gema Chamorro-Moriana**[1,2]*, **Fernando Espuny-Ruiz**[1]*, **Carmen Ridao-Fernández**[1,2], **Eleonora Magni**[3]

**1** Department of Physiotherapy, Faculty of Nursing, Physiotherapy and Podiatry, University of Seville, Seville, Spain, **2** Research Group "Area of Physiotherapy CTS305", Spain, **3** Department of Nursing, Faculty of Nursing, Physiotherapy and Podiatry, University of Seville, Seville, Spain

* f.espuny_98@hotmail.com (FER); gchamorro@us.es (GCM)

## Abstract

### Objectives

To determine the most appropriate method of functional assessment for "patellofemoral pain" (PFP)/"chondromalacia patella" for its diagnostic value, (validity, reliability, sensitivity, specificity, predictive value and clinical applicability); to outline initial interpretations of the questionnaires and their appropriateness, through the cut-off points determined in their scores based on physical test and Magnetic Resonance Imaging (MRI); to establish which methods should be used in conjunction with each other to obtain clinical diagnoses that are robust effective and efficient.

### Methods

(1)Intra- and inter-observer reliability and of the relationship among PFP questionnaires/ physical tests validated. (2)Predictive capacity of the questionnaires. Subject: 113 knees with PFP, assessed using "*Knee-injury-and-Osteoarthritis Outcome-Score-for-Patellofe-moral-pain-and-osteoarthritis*" (KOOS-PF), "*Kujala-Patellofemoral-Score*" (KPS), "*Victo-rian-Institute-of-Sports-Assessment-for-Patellar-tendons-questionnaire*" (VISA-P), and the physical tests: "*patellar-palpation*", "*patellar-tilt*", "*patellar-apprehension*", "*Clarke*" and "*squat*".

### Results

Questionnaires correlations themselves was $0.78 < r < 0.86$. Tests intra-rater reliability was "*excellent*". *Squat* inter-rater reliability was "*excellent*"/"*good*". *Palpation*, *tilt*, *Clarke* and *squat* showed a statistically significant relationship ($p < 0.05$) with all questionnaires/specific items. AUC of the questionnaires showed a "useful" accuracy, except for *Tilt*. No statistically significant differences were found between grades 0 and 1 chondromalacia (by MRI) knee scores, but between 1 and $\geq 2$. AUC of the questionnaires showed "useful" accuracy.

**Data Availability Statement:** All relevant data are within the manuscript and its Supporting Information files.

**Funding:** The author(s) received no specific funding for this work.

**Competing interests:** The authors have declared that no competing interests exist.

## Conclusions

KOOS-PF, KPS and VISA-P demonstrated their diagnostic value in PFP/chondromalacia (validity, reliability, sensitivity, specificity, predictive value and clinical applicability). KOOS-PF was the most versatile, and the most appropriate in mild cases and for early detection and prevention. *Squat* was the best due to its reliability and clinical relationship with the questionnaires, which predicted it correctly. The functional assessment tools discussed should be applied by combining them with each other.

## Introduction

Patellofemoral Pain (PFP), also known as Patellofemoral Pain Syndrome [1], is a condition characterized by retropatellar or peripatellar pain [1–3] often of insidious onset [4] associated primarily with activities where the support of body weight in knee flexion positions loads the patellofemoral joint [2, 3]. It is often considered a chronic overuse injury [5], with symptoms persisting for up to 20 years [6] leading to a diminished quality of life [7]. Chondromalacia patella (CP) can be used as a PFP synonym if there is structural cartilage damage [8] diagnosed by imaging [9], or simply grade I cartilage softening and inflammation, according to several commonly used classifications, such as Outerbridge [10], Insall [11], Noyes [11] or even Modified Noyes [12].

It has a high prevalence, especially among the military, adolescents and athletes in general [5]. For example, in 2016, it affected 22.7% of the UK population [5] although some authors estimate this figure at over 40% [13, 14]. Women are twice as likely to suffer from PFP [5], possibly due to anatomical and biomechanical variations, including thinner cartilage, greater Q-angle, greater dynamic valgus [4, 15], etc.

The pathogenesis of PFP is multifactorial [15, 16]. Thus, quadriceps weakness or quadriceps muscle imbalance [15, 16], patellar instability [1], foot or hip dysfunction [15], activities such as running, squats or stair climbing [6], and dynamic valgus in women [15], among other factors, lead to incorrect patellar tracking [15], hypermobile [1, 15] or lateralised patellae [15], etc. This, together with the influence of abnormal morphological features [17, 18] regarding patellar type [19], sulcus depth of femoral condyles [17–19], patellar tilt angles [17], etc., results in ongoing patellofemoral friction that contributes or predisposes to the onset of pain and even cartilage deterioration [18].

The diverse etiopathology must be taken into account in the assessment and subsequent medical and physiotherapeutic process [20]. Numerous methods have been used to assess PFP [21]. For example, imaging is useful to objectively confirm whether there is structural damage to the cartilage [9]. Magnetic Resonance Imaging (MRI), which appears to be accurate in high grades [22], and arthroscopies; show low correlation between degrees and symptoms [22]. On the other hand, MRI, computed tomography, ultrasound and X-ray are useful for locating the patella in different degrees of flexion, although the first three are only used in unloaded flexion [23]. Thus, they provide interesting but incomplete data based on the preceding paragraph. Non-technological methods, i.e. functional assessment questionnaires and physical tests [21, 24, 25], stand out for their efficiency, clinical feasibility and focus mainly on symptomatology. Few questionnaires are validated and created specifically for these patellofemoral conditions, i.e. Kujala Patellofemoral Score (KPS) [26]. Since cross-cultural language adaptations are not always available or they are not accessible to researchers and clinicians, among others, non-

specific questionnaires are also frequently used to assess PFP [20, 21, 27]. This is the case of Fulkerson Knee Instability Scale (FKIS) [28].

KPS is considered the reference standard, among other reasons because of how long it has been in use (since 1993) and its methodological quality [29, 30]. The most common physical tests are: *medial/lateral patellar palpation* [2, 9, 31, 32], *Clarke test* [2, 9, 31, 32] as specific palpation and mobility tests, and the *squat test* [2, 9, 32, 33] as a general pain provocation test. Up to now, it has not been possible to identify a benchmark physical test, which is needed [31, 32] in clinical and research settings in order to apply it in isolation or in combination with other tests [34].

The difficulty involved in diagnosing PFP due to its multifactorial nature [15, 16] and the differential diagnoses it entails, such as bursitis, patellar tendinopathy or rheumatoid arthritis [35], lead to an in-depth examination of the efficiency of existing assessment methods and the identification of clinically feasible tools. This could help to improve assessment [25, 33], especially early ones, and consequently therapeutic protocols, whether preventive or curative. The high prevalence of patellofemoral conditions [5], the poor long-term prognosis [1, 4, 6, 36] due to cartilage degeneration [18], its association with disability and chronicity [6, 7], and the economic and health care costs involved [5] are other issues that would justify the need to further develop the existing assessment methods for PFP.

Since, so far, there is no evidence in the literature of a functional assessment method that provides all of the above-mentioned characteristics and is also useful especially in initial; this study aimed to determine the most appropriate method of functional assessment to evaluate "patellofemoral pain" and/or "CP" for its diagnostic value, according to its validity, reliability, sensitivity, specificity, predictive value and clinical applicability. A second objective was to outline initial interpretations of the questionnaires and their appropriateness, through the cut-off points determined in their scores based on physical and MRI tests. Lastly, this study is aimed to establish which methods should be used in conjunction with each other in order to obtain clinical diagnoses that are robust, effective and efficient.

## Materials and methods

Study 1: intra- and inter-observer reliability and of the relationship among PFP functional assessment methods: questionnaires validated and/or applied in PFP and cross-culturally adapted to Spanish, as well as physical tests validated in PFP. Study 2: predictive capacity of the above-mentioned questionnaires.

### Participants and sample

Participants were recruited using non-random convenience sampling. Inclusion criteria were: subjects with symptomatic PFP with or without cartilage damage (note: by PFP we mean the patellofemoral joint conflict that produces friction between the facet joints due to non-traumatic causes such as knee valgus, muscle imbalance between the vastus medialis and vastus lateralis, etc., as shown in the introduction section); between 16 and 55 years old to avoid late symptoms of apophysitis (Osgood-Schlatter or Sinding-Larssen-Johansson) and early symptoms of osteoarthritis [37]; and native Spanish speakers. Exclusion criteria were: subjects with severe cognitive or coordination impairment, severe cardiovascular or respiratory conditions and PFP symptomatology common to other affected knee dysfunction such as femoral condyle or patella fractures affecting the facet joints, knee prosthesis, etc.; and disorders of other joints such as hip, ankle or femorotibial joint, which prevent the performance of physical tests or bias the tests results.

The study considered the affected knees as the sample, except for the descriptives, where it was the participants.

## Functional assessment methods used

Following a literature review using Pubmed, Web of Science, Scopus, Cinahl and Dialnet, we selected three validated self-administered functional assessment questionnaires, transculturally adapted to Spanish and applied to PFP. Knee injury and Osteoarthritis Outcome Score for Patellofemoral pain and osteoarthritis (KOOS-PF), which assesses PFP and/or osteoarthritis [30], addressing pain especially, stiffness, and quality of life. KPS, which assesses patellofemoral pathologies [26] considering pain and physical alterations, functional limitation and difficulty in sports activities. Victorian Institute of Sports Assessment for Patellar tendons questionnaire (VISA-P), which is applied in PFP [29], although it was designed specifically for patellar tendinopathy [38]. It especially considers pain and the ability to engage in physical activity and sport. The scores of the three questionnaires ranged 0–100 (optimal/asymptomatic).

The regular functional tests selected, all of which were dichotomous (positive/negative), were: *patellar palpation test* [31], positive if tenderness or discomfort is present in any of the medial or lateral facets; *patellar tilt test* [31], positive if the lateral outer edge of the patella is not elevated or does not separate from the femur; *patellar apprehension test* [31], positive if there is a withdrawal manoeuvre by the patient when the patella is displaced outwards; *Clarke test*, positive if there is pain on isometric contraction of the quadriceps and cranial displacement of the patella rubbing against the femur or if the contraction is avoided for fear of pain [39]; and the *squat test* with 90˚ of knee flexion, positive if it causes pain or if pain prevents them from performing the test [40].

## Magnetic Resonance Imaging (MRI)

MRI has been used in this study to establish cut-off points in the questionnaire scores according to the CP degrees it determines. The degrees [10] are: I, cartilage softening and swelling; II, fissures <0.5 inches in diameter not reaching the subchondral bone; III, fissures >0.5 inches reaching the subchondral bone; and IV, erosion that exposes subchondral bone.

## Action protocol

Evaluations were conducted between 01 December 2022 and 28 February 2023. Each subject was assessed in two sessions, 7–10 days apart; enough time to avoid recall of their answers, but not too long after to avoid clinical changes in PFP [29, 41], since the purpose of the study was not to assess the effect of an intervention.

Session 1. 1st Completion of the informed consent form and collection of descriptive data on the subject, both general information and regarding PFP (affected knee, summary of the process and Q-angle assessment of the affected knee(s). 2nd Self-completion of questionnaires in the following order: "KOOS-PF", "KPS", "VISA-P". They were able to ask the investigators any questions they had. 3rd Application of physical assessment tests by two physiotherapists on two alternative occasions (FE-CR-FE-CR) in the following order: *patellar palpation*, *patellar tilt*, *patellar apprehension*, *Clarke* and *squat*. Since the last two could cause pain, they were left for last to avoid apprehension regarding the other tests [39].

Session 2. 1st Self-completion of the questionnaires as in session 1. 2nd Battery of physical tests only performed by a physiotherapist (FE-FE).

## Statistical analysis

Sample size calculation was based on a related study that examined the combination of history elements and physical tests for PFP [25]. The following conditions were used: KOOS-PF mean standard deviations (*s*), 20.22; confidence interval of 95% and error (*E*), 4. The following formula used was:

$$n = \frac{z_{\alpha/2}^2 \cdot s^2}{E^2}$$

The sample size obtained was 99 affected knees. Our sample included a few more subjects following COSMIN recommendations [42] (optimum $\geq$ 100) and anticipating possible dropouts.

The description of the participants and their condition considered absolute (N) and relative (%) frequencies in qualitative variables. The normality of quantitative variables was analysed using the Shapiro-Wilk test, considering mean and standard deviation for parametric variables, and median and interquartile range for non-parametric variables.

To assess the relationship between the two measurement moments of the questionnaires, we used: the t-Test for related samples in normal distributions, and the Wilcoxon Signed-Rank Test for related samples in non-normal distributions. We applied a double correlation analysis among questionnaires, one for each session (time). Pearson's correlation coefficient (*r*) and Spearman's Rho were used, depending on whether the distribution was normal or not, respectively, where: $r > 0.7$ strong, $0.7 \geq r > 0.5$ moderate, $0.5 \geq r > 0.25$ weak and $r \leq 0.25$ rare correlation [43]. The same correlation analysis was applied among some specific items of the questionnaires due to their direct link to the *squat test*.

The inter-observer and intra-observer agreement of the tests was analyzed using Cohen's Kappa Index (*k*). Inter-observer reliability was analyzed twice: between the 1st FE and CR measurements (session 1); and between the 2nd FE and CR measurements (session 1). Intra-observer reliability was analyzed twice: between the 1st and 2nd FE measurements (session 1) and between the next 2 FE measurements (session 2). It was graded according to Landis JR et al [44] where: $k > 0.8$ excellent, $0.8 \geq k > 0.6$ good, $0.6 \geq k > 0.4$ acceptable and $k \leq 0.4$ unacceptable.

The relationships among questionnaires and physical tests were analyzed with t-Test for independent samples in variables with normal distributions, and Mann-Whitney U-test for independent samples in non-normal distributions. The 2 measurement moments of the questionnaires and the physical tests were taken into account. Notice that only those test measurements where the positive/negative ratio $\neq 1$ at each point in time were considered for the comparative analysis. For each questionnaire the following was calculated: area under curve (AUC) (between 0–1), represented by a ROC curve, cut-off point to discriminate between positive and negative results in each physical test, sensitivity, specificity, positive predictive value (PPV) and negative predictive value (NPV) (CI = 95%). AUC was graded according to Swets JA [45] where: 0.5–0.7 = "low" accuracy, 0.7–0.9 = "useful" accuracy, >0.9 = "high" accuracy. The ROC curve shows the probability of the questionnaire being able to discriminate between positive and negative physical test results.

On the other hand, the same statistical tests mentioned in the previous paragraph were carried out for some specific items associated with the *patellar tilt test*, *Clarke test* and *squat test*. Similarly, a comparative analysis between CP grades (0, 1 and $\geq$1) by MRI and questionnaire scores was carried out with a sub-sample of the study. Cluster analysis was performed to establish groupings in the sample based on overall questionnaire scores, physical test results, gender, height and Q-angle. Moreover, this analysis will make it possible to establish the relationships between the variables of the study based on the association between gender, height and Q-

angle. Only results of cohesion and separation silhouette measurement >0.5 (<0.2 = "poor"; 0.2–0.5 = "sufficient"; >0.5 = "good") were considered valid [46].

$p<0.05$ values were considered statistically significant in general. CI95%>0.5 were considered statistically significant for AUC [47].

IBM SPSS STATISTICS 25® software was used for the analysis.

This research was conducted in accordance with the Declaration of Helsinki and approved on 3 December 2021 by the Research Ethics Committee of the Virgen Macarena-Virgen del Rocío Hospitals of the Andalusian Public Health System (C.I.0162-N-21). Before starting the fieldwork (request for personal data and functional assessment), all participants received a written information sheet about the study together with the informed consent to sign. They were able to ask questions and solve any doubts about these documents with the researchers before signing the consent, which did not prevent the subject from leaving the study if he/she wanted to. All patients gave written informed consent after reading the user information sheet. For those subjects under 18 (inclusion criterion: minimum 16 years), the consent signatures of the parent or legal guardian and the informed assent of the minor, i.e. in age-appropriate language, were expressly required. All the above was included in the application submitted to the Ethics Committee for approval.

## Results

A total of 113 affected knees of 80 participants were assessed. The description of both variables is detailed in Table 1.

### Analysis of assessment questionnaires

The relationship between the measurement moments of the assessment questionnaires is shown in Table 2. Bilaterally affected participants answered for each of their affected knees in

**Table 1. Descriptive characteristics of the participants and sample.**

| | | | Men (n = 29) | Women (n = 51) | Total (n = 80) |
|---|---|---|---|---|---|
| **Age (years)** | | **Med** | 30.5 | 29 | 30 |
| | | **IQR** | 22.3–49.0 | 23.0–48.5 | 23–49 |
| **Height (cm)** | | **μ** | 181 | 166 | 172 |
| | | **SD** | 6 | 6 | 9 |
| **BMI** | | **Med** | 24.7 | 23 | 24.4 |
| | | **IQR** | 23.5–29.2 | 20.8–26.1 | 22.0–27.9 |
| **Time with pain (months)** | | **Med** | 66 | 54 | 60 |
| | | **IQR** | 27–120 | 24–84 | 24–99 |
| **Impairment** | *Bilateral* | **N** | 11 | 22 | 33 |
| | | **%** | 14% | 27.5% | 41.5% |
| | *Unilateral* | **N** | 18 | 29 | 47 |
| | | **%** | 22.5% | 36% | 58.5% |
| **Impaired limb\*** | *Dominant* | **N** | 11 | 17 | 28 |
| | | **%** | 24% | 36% | 60% |
| | *Non- dominant* | **N** | 7 | 12 | 19 |
| | | **%** | 15% | 25% | 40% |
| **Affected knees Q-angle (degrees)** | | **Med** | 16 | 19 | 18 |
| | | **IQR** | 14–18 | 17–24 | 16–22 |

Abbreviations: BMI, body mass index; Med, median; IQR, interquartile range.

[a] Only unilateral affected subjects considered.

**Table 2. Relationship between the two measurement moments of the rating questionnaires.**

|  |  | μ | SD | Med | IQR | p |
|---|---|---|---|---|---|---|
| **KOOS-PF** | **m1** | 54.5 | 22.8 | 59.1 | 34.1–69.4 | <0.001* |
|  | **m2** | 60.5 | 23.0 | 63.6 | 47.7–77.3 |  |
| **KPS** | **m1** | 73.5 | 17.6 | 78 | 62.0–86.5 | 0.078* |
|  | **m2** | 74.7 | 18.3 | 80 | 65–89 |  |
| **VISA-P** | **m1** | 55.6 | 21.3 | 56 | 41–73 | 0.249[a] |
|  | **m2** | 56.8 | 22.5 | 57 | 41–73 |  |

Abbreviations: m1, measurement 1; m2, measurement 2; μ, media; SD, standard deviation; Med, median; IQR, interquartile range.

[a] Wilcoxon signed-rank test for paired samples.

[b] T-test for paired samples.

Questionnaires score ranges: 0–100.

the questions that required them to do so. The sample in the following tables was thus the knees, not the participants.

Statistically significant differences were found in KOOS-PF.

Table 3 shows the correlation of the rating questionnaires with each other at the two measurement moments.

The 3 questionnaires showed statistically significant correlation among them ($p<0.05$) at the two moments in time analyzed. In fact, all were $p<0.001$, their correlation coefficients being >0.78.

## Analysis of physical assessment tests

The intra-observer and inter-observer agreement analyses of the different physical tests are shown in Table 4.

Summarizing the table above, the inter-observer reliability of the tests analyzed showed results ranging from "excellent" ($k>0.8$) [44] to "acceptable" ($0.4<k<0.6$) [44]. The *squat test* stood out with an "excellent" ($k>0.8$) [44] result in the first measurement, and "good" ($0.6<k<0.8$) [44] in the second.

In terms of intra-observer reliability, the analyses of all tests produced "excellent" ($k<0.8$) [44] results.

## Analysis of the relationships among questionnaires and physical assessment tests

The relationship among questionnaires and tests is summarized in Tables 5 and 6. Table 5 shows whether subjects with scores on the questionnaires referring to milder pathology were more likely to have a negative test result (absence of symptoms) and viceversa. Table 6 shows

**Table 3. Correlation among rating questionnaires at the two measurement moments.**

| r | Measurement 1 | | Measurement 2 | |
|---|---|---|---|---|
|  | **KOOS-PF** | **KPS** | **KOOS-PF** | **KPS** |
| **KPS** | 0.825 |  | 0.865 |  |
| **VISA-P** | 0.851 | 0.788 | 0.860 | 0.819 |

Abbreviation: r, correlation coefficient.

Note: all coefficients were analyzed using Spearman's Rho. All values were $p<0.001$.

**Table 4. Reliability of physical assessment tests.**

| Physical tests | Inter-rater | | Intra-rater | |
|---|---|---|---|---|
| | *1FE-1CR* | *2FE-2CR* | *1FE-2FE* | *3FE-4FE* |
| **Patellar palpation** | 0.619 | 0.697 | 0.809 | 0.942 |
| **Patellar tilt** | 0.505 | 0.540 | 0.826 | 0.954 |
| **Clarke** | 0.600 | 0.554 | 0.849 | 0.858 |
| **Squat** | 0.862 | 0.732 | 0.904 | 0.903 |

Abbreviations: 1FE, 1st measurement FE; 2FE, 2nd measurement FE; 3F, 3rd measurement FE; 4F, 4th measurement FE; 1CR, 1st measurement CR; 2CR, 2nd measurement CR.

the capacity of the questionnaires and some of their specific items to predict test outcome, and the cut-off points of the questionnaire scores from which they are predicted. *Patellar apprehension test* is not reflected in the following tables because all test measurements were negative.

*Patellar palpation test*, *patellar tilt test*, *Clarke test* and *squat test* showed a statistically significant relationship ($p < 0.05$) with all 3 questionnaires and specific items, i.e. positive tests corresponded to lower scores (higher in KOOS-PF items). All values were $p < 0.001$, except for *tilt test*.

The AUC values of the 3 questionnaires were significant (CI95% > 0.5) showing a "useful" accuracy (0.7–0.9) [45], except for the *patellar tilt test*.

Associations with the squat test (> 0.8) were notable. The AUC values of the 7 specific items analysed, also significant, indicated a "useful" accuracy (0.7–0.9) [45], except KPS-4 with *squat*.

The representation of the ROC curves is shown in Fig 1.

All CI95% of the sensitivity, specificity, PPV and NPV of the three questionnaires with the *squat* were > 50%.

The CI95% for the sensitivity of the 7 specific items were > 50%, although none scored > 50% for all statistics.

Table 7 shows the correlation between the specific items associated with the *squat test*.

All items associated with the *squat test* showed statistically significant correlations ($p < 0.001$).

## Analysis of the relationships among questionnaries and CP degree according to MRI

The relationship among questionnaires and CP is summarized in Tables 8 and 9. Table 8 shows whether subjects with higher scores in the questionnaires (milder pathology), were more likely to have lower degrees of CP (less structural damage). Table 9 shows the ability of the questionnaires to predict the degree of CP and the cut-off points of the questionnaire scores from which they are predicted.

No statistically significant differences were found between grade 0 and grade 1 CP knee scores, but there were differences between grade 1 and ≥2 grades.

All AUC values for the 3 questionnaires were statistically significant and of "useful" accuracy (0.7–0.9) [45]. The representation of the ROC curves is shown in Fig 2.

## Cluster analysis

Finally, the sample clusters are shown by cluster analysis in relation to the questionnaires, gender, height and Q-angle. Note: all physical tests were discarded by the statistical test itself. The optimal algorithm generated presented 3 clusters with 6 entries, in agreement with the good

**Table 5. Descriptive and comparative analysis of the overall scores of the questionnaires and specific items according to the results of the physical tests.**

| | | Physical tests | | N (knees) | μ | SD | Med | IQR | P |
|---|---|---|---|---|---|---|---|---|---|
| Questionnaires | KOOS-PF | Patellar palpation | - | 82 | 69.4 | 19.1 | 68.2 | 61.4–84.7 | <0.001[a] |
| | | | + | 142 | 50.6 | 22.5 | 55.7 | 34.1–68.2 | |
| | | Patellar tilt | - | 58 | 64.2 | 22.3 | 68.2 | 46.0–77.9 | 0.007[a] |
| | | | + | 156 | 54.9 | 22.7 | 59 | 37.0–72.7 | |
| | | Clarke | - | 126 | 67.0 | 18.0 | 68.2 | 56.2–77.9 | <0.001[a] |
| | | | + | 80 | 42.0 | 22.5 | 37.5 | 21.1–61.4 | |
| | | Squat | - | 148 | 67.0 | 17.3 | 68.2 | 56.8–77.3 | <0.001[a] |
| | | | + | 76 | 39.1 | 21.8 | 34.1 | 20.5–56.8 | |
| | KPS | Patellar palpation | - | 82 | 81.7 | 13.8 | 85 | 76–91 | <0.001[a] |
| | | | + | 142 | 69.6 | 18.7 | 76 | 53.8–83.0 | |
| | | Patellar tilt | - | 58 | 79.2 | 16.1 | 84 | 67.8–91.3 | 0.004[a] |
| | | | + | 156 | 72.2 | 18.3 | 78 | 62–85 | |
| | | Clarke | - | 126 | 81.5 | 12.1 | 83 | 74–91 | <0.001[a] |
| | | | + | 80 | 61.7 | 19.4 | 62 | 43.0–80.8 | |
| | | Squat | - | 148 | 81.0 | 13.5 | 82 | 76–91 | <0.001[a] |
| | | | + | 76 | 60.5 | 17.9 | 61 | 43–78 | |
| | VISA-P | Patellar palpation | - | 82 | 67.4 | 18.2 | 66.5 | 53–83 | <0.001[b] |
| | | | + | 142 | 49.6 | 21.3 | 51 | 34–64 | |
| | | Patellar tilt | - | 58 | 62.5 | 23.1 | 60.5 | 43.5–85.0 | 0.031[a] |
| | | | + | 156 | 54.0 | 21.1 | 55 | 38.3–69.8 | |
| | | Clarke | - | 126 | 64.8 | 17.6 | 64.5 | 51–78 | <0.001[b] |
| | | | + | 80 | 42.7 | 20.7 | 42 | 27.0–56.8 | |
| | | Squat | - | 148 | 64.9 | 18.3 | 63.5 | 53.3–78.8 | <0.001[b] |
| | | | + | 76 | 39.1 | 18.4 | 38.5 | 25.3–52.0 | |
| Specific items | KPS-13 | Patellar tilt | - | 58 | 4.1 | 1.4 | 5 | 3–5 | 0.004[a] |
| | | | + | 156 | 3.8 | 1.3 | 3 | 3–5 | |
| | VISA-P-3 | Clarke | - | 126 | 8.6 | 1.8 | 9 | 8–10 | <0.001[a] |
| | | | + | 80 | 6.2 | 2.6 | 6 | 4.0–8.8 | |
| | KOOS-PF-4 | Squat | - | 148 | 1.1 | 0.9 | 1 | 0–2 | <0.001[a] |
| | | | + | 76 | 1.9 | 1.1 | 2 | 1–3 | |
| | KOOS-PF-6 | | - | 148 | 1.5 | 1.1 | 1 | 1–2 | <0.001[a] |
| | | | + | 76 | 2.7 | 1.1 | 3 | 2–4 | |
| | KPS-4 | | - | 148 | 7.9 | 1.8 | 8 | 8–10 | <0.001[a] |
| | | | + | 76 | 6.6 | 1.8 | 5 | 5–8 | |
| | KPS-5 | | - | 148 | 3.7 | 0.8 | 4 | 3–4 | <0.001[a] |
| | | | + | 76 | 2.8 | 1.3 | 3 | 2–4 | |
| | VISA-P-2 | | - | 148 | 8.6 | 1.8 | 9 | 8–10 | <0.001[a] |
| | | | + | 76 | 6.1 | 2.7 | 6 | 4–9 | |
| | VISA-P-5 | | - | 148 | 7.6 | 2.3 | 8 | 7–10 | <0.001[a] |
| | | | + | 76 | 6.3 | 2.6 | 7 | 4–8 | |

Abbreviations: μ, media; SD, standard deviation; Med, median; IQR, interquartile range.

\* The + and−signs indicate the results of the physical test.

\*\* The *p* value indicates the statistical significance of the dependence between the test result and the scale/item scores. The greater the difference in scores among those scales that correspond to positive test results and those that correspond to negative test results, the greater the statistical significance.

[a] Mann-Whitney U test for non-paired samples.

[b] T-test for non-paired samples.

Items: KPS-13, "*Flexion deficiency*"; VISA-P-3, "*Do you have pain at the knee with full active non weight bearing knee extension*?"; KOOS-PF-4, "*Rising from sitting*"; KOOS-PF-6, "*Squatting*"; KPS-4, "Stairs"; KPS-5, "*Squatting*"; VISA-P-2, "*Do you have pain walking down stairs with a normal gait cycle*?"; VISA-P-5, "*Do you have problems squatting*?*".

Score ranges: questionnaires (0–100); KOOS-PF items (0–4); KPS-4 (0–10); KPS-5 and KPS-13 (0–5); VISA-P items (0–10).

**Table 6. Predictive ability of questionnaires and specific items on the results of physical tests.**

| | | Physical test | | Cut-off | AUC | CI95% | Sens (%) | CI95% | Spec (%) | CI95% | PPV (%) | CI95% | NPV (%) | CI95% |
|---|---|---|---|---|---|---|---|---|---|---|---|---|---|---|
| Questionnaires | KOOS-PF | Patellar palpation | - | 60.25 | 0.739 | 0.672–0.806 | 76.8 | 66.6–84.6 | 61.3 | 53.1–68.9 | 53.4 | 44.4–62.1* | 82.1 | 73.7–88.2 |
| | | Patellar tilt | - | 60.25 | 0.620 | 0.534–0.707 | 69.0 | 56.2–79.4 | 53.2 | 45.4–60.9* | 35.4* | 27.2–44.6* | 82.2 | 73.6–88.4 |
| | | Clarke | - | 48.85 | 0.798 | 0.737–0.859 | 83.3 | 75.9–88.8 | 56.3 | 45.3–66.6* | 75.0 | 67.2–81.4 | 68.2 | 56.2–78.2 |
| | | Squat | - | 51.15 | 0.831 | 0.772–0.890 | 83.8 | 77.0–88.9 | 71.1 | 60.0–80.0 | 84.9 | 78.2–89.8 | 69.2 | 58.3–78.4 |
| | KPS | Patellar palpation | - | 79.5 | 0.704 | 0.634–0.774 | 70.7 | 60.1–79.5 | 63.4 | 55.2–70.9 | 52.7 | 43.5–61.8* | 78.9 | 70.6–85.4 |
| | | Patellar tilt | - | 79.5 | 0.628 | 0.541–0.714 | 63.8 | 50.9–74.9 | 57.7 | 49.8–65.2* | 35.9* | 27.3–45.5* | 81.1 | 72.8–87.3 |
| | | Clarke | - | 72.5 | 0.793 | 0.731–0.855 | 81.0 | 73.2–86.9 | 58.8 | 47.8–68.9* | 75.6 | 67.7–82.0 | 66.2 | 54.6–76.1 |
| | | Squat | - | 72.5 | 0.818 | 0.760–0.876 | 82.4 | 75.5–87.7 | 64.5 | 53.3–74.3 | 81.9 | 74.9–87.2 | 65.3 | 54.1–75.1 |
| | VISA-P | Patellar palpation | - | 54.5 | 0.730 | 0.664–0.795 | 73.2 | 62.7–81.6 | 56.3 | 48.1–64.2* | 49.2* | 40.5–57.9* | 78.4 | 69.5–85.3 |
| | | Patellar tilt | - | 55.5 | 0.596 | 0.507–0.685 | 60.3 | 47.5–71.9* | 50.6 | 42.9–58.4* | 31.3* | 23.4–40.3* | 77.5 | 68.4–84.5 |
| | | Clarke | - | 48.5 | 0.791 | 0.726–0.855 | 80.2 | 72.3–86.2 | 60.0 | 49.0–70.0* | 75.9 | 68.0–82.4 | 65.8 | 54.3–75.6 |
| | | Squat | - | 45.5 | 0.838 | 0.784–0.893 | 85.1 | 78.5–90.0 | 63.2 | 51.9–73.1 | 81.8 | 75.0–87.1 | 68.6 | 57.0–78.2 |
| Specific items | VISA-P-3 | Clarke | - | 6.5 | 0.782 | 0.718–0.846 | 88.1 | 81.3–92.7 | 53.8 | 42.9–64.3* | 75.0 | 67.5–81.3 | 74.1 | 61.6–83.7 |
| | KOOS-PF-4 | Squat | + | 0.5 | 0.716 | 0.643–0.789 | 89.5 | 80.6–94.6 | 29.1* | 22.3–36.8* | 39.3* | 32.3–46.7* | 84.3 | 72.0–91.8 |
| | KOOS-PF-6 | | + | 1.5 | 0.780 | 0.717–0.844 | 81.6 | 71.4–88.7 | 58.8 | 50.7–66.4 | 50.4 | 41.7–59.1* | 86.1 | 78.1–91.6 |
| | KPS-4 | | - | 6.5 | 0.686 | 0.613–0.759 | 77.0 | 69.6–83.1 | 53.9 | 42.8–64.7* | 76.5 | 69.1–82.6 | 54.7 | 43.4–65.4* |
| | KPS-5 | | - | 2.5 | 0.731 | 0.661–0.800 | 94.6 | 89.7–97.2 | 26.3* | 17.7–37.2* | 71.4 | 64.7–77.3 | 71.4 | 52.9–84.7 |
| | VISA-P-2 | | - | 6.5 | 0.749 | 0.678–0.820 | 77.0 | 69.6–83.1 | 64.5 | 53.3–74.3 | 80.9 | 73.6–86.5 | 59.0 | 48.3–69.0* |
| | VISA-P-5 | | - | 2.5 | 0.799 | 0.734–0.863 | 90.5 | 84.7–94.3 | 57.9 | 46.7–68.4* | 80.7 | 74.1–86.0 | 75.9 | 63.5–85.0 |

Abbreviations: AUC, Area Under Curve; CI, Confidence Interval; Sens, Sensitivity; Spec, Specificity; PPV, Positive Predictive Value; NPV, Negative Predictive Value.

a Values ≤50%

* + and–signs indicate the result of the test that are analyzed. Therefore, the table indicates that those questionnaires/items that score higher than the cut-off indicate a higher probability of a negative test result (in KOOS-PF-4 and KOOS-PF-6 would indicate a negative test result).

Items: VISA-P-3, "*Do you have pain at the knee with full active non weight bearing knee extension*?"; KOOS-PF-4, "*Rising from sitting*"; KOOS-PF-6, "*Squatting*"; KPS-4, "*Stairs*"; KPS-5, "*Squatting*"; VISA-P-2, "*Do you have pain walking down stairs with a normal gait cycle*?"; VISA-P-5, "*Do you have problems squatting*?*".

Score ranges: questionnaires (0–100); KOOS-PF items (0–4), KPS-4 (0–10), KPS-5 (0–5), VISA-P items (0–10).

Notes: all AUC CI95% values were >0.5 except for item KPS-13 (not included) related to *patellar tilt test*.

cluster quality achieved through the silhouette measure of cohesion and separation, with a value of 0.5213 (Fig 3). Fig 4 shows the clusters.

Regarding the median of the total sample, cluster 1 includes higher men with lower Q-angles and questionnaires with higher scores.

Clusters 2 and 3 group women: cluster 2 with heights below the median, widely larger Q-angles and lower-scoring questionnaires; and cluster 3 with heights close to the median, slightly larger Q-angles and higher-scoring questionnaires.

## Discussion

This study analyzed the relationship among the functional assessment methods frequently used in PFP subjects, both in clinical and research settings: KOOS-PF, KPS and VISA-P questionnaires; and physical tests *patellar palpation*, *patellar tilt*, *patellar apprehension*, *Clarke* and *squat*. Among the most important findings were the strong correlation among the questionnaires, the "excellent" intra-observer reliability of the physical tests and the "excellent" and

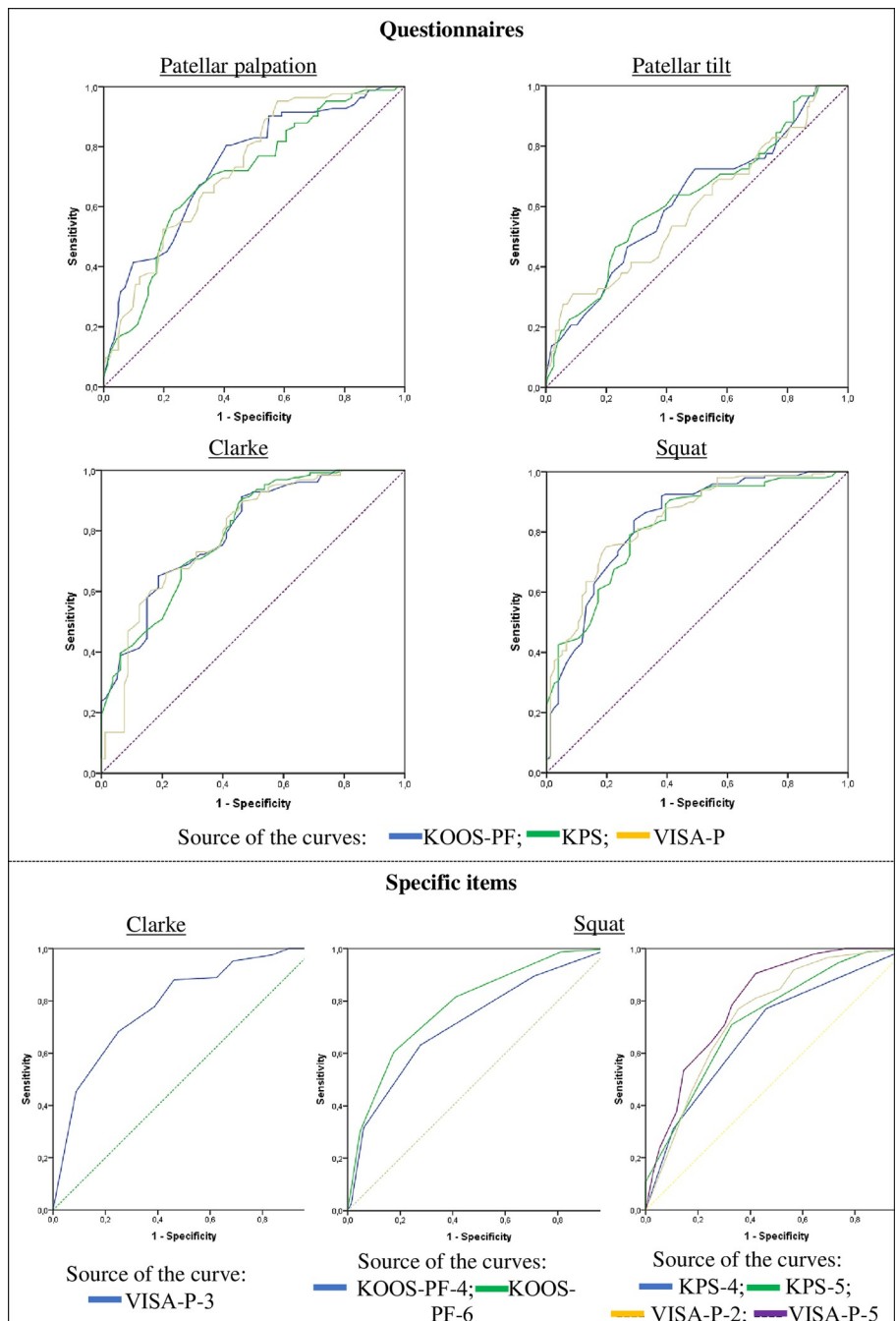

**Fig 1. ROC curves of the questionnaires in general and of the specific items associated with the *Clarke* and *squat* tests.** \*\*Diagonal segments are generated by ties. Items: VISA-P-3, "*Do you have pain at the knee with full active non weight bearing knee extension*?"; KOOS-PF-4, "*Rising from sitting*"; KOOS-PF-6, "*Squatting*"; KPS-4, "Stairs"; KPS-5, "*Squatting*"; VISA-P-2, "*Do you have pain walking down stairs with a normal gait cycle*?"; VISA-P-5, "*Do you have problems squatting*?*"*. Note: reference lines are marked with dashes.

"good" inter-observer reliability of the *squat*. Also of interest is the relationship between *patellar palpation*, *patellar tilt*, *Clarke* and *squat* and the questionnaires. In relation to the predictive capacity of the questionnaires, it was evidenced regarding the physical tests mentioned, especially with *squat*; and the degrees of CP by MRI.

**Table 7. Correlation among specific items linked to *squat test*.**

| *r* | KOOS-PF-4 | KOOS-PF-6 | KPS-4 | KPS-5 | VISA-P-2 |
|---|---|---|---|---|---|
| **KOOS-PF-6** | 0.440 | | | | |
| **KPS-4** | -0.506 | -0.472 | | | |
| **KPS-5** | -0.395 | -0.497 | -0.470 | | |
| **VISA-P-2** | -0.513 | -0.567 | -0.654 | -0.527 | |
| **VISA-P-5** | -0.517 | -0.658 | -0.518 | -0.644 | -0.652 |

Abbreviation: *r*, correlation coefficient.

Items: KOOS-PF-4, "*Rising from sitting*"; KOOS-PF-6, "*Squatting*"; KPS-4, "Stairs"; KPS-5, "*Squatting*"; VISA-P-2, "*Do you have pain walking down stairs with a normal gait cycle*?"; VISA-P-5, "*Do you have problems squatting*?".

Notes: all coefficients were analyzed using Spearman's Rho. All values were *p*<0.001.

KOOS-PF and VISA-P were better predictors of grades greater than 1. Cluster analysis logically associated questionnaire scores, gender, height and Q-angle.

Thus, taller women are associated with higher Q-angle and healthier questionnaire scores. These and other results are discussed below.

## About the assessment questionnaires

In relation to the different times at which participants completed the questionnaires, no statistically significant differences were found between the KPS and VISA-P scores. However, there were discrepancies regarding the KOOS-PF measurements, possibly due to addressing symptomatology in the last week. This implied that some sporadic activity prior to the assessment may have influenced the subject's response [48].

**Table 8. Comparative analyses of questionnaire scores according to the degree of chondromalacia patella (CP).**

| | Degree | N | μ | SD | Med | IQR | P |
|---|---|---|---|---|---|---|---|
| **KOOS-PF** | 0 | 20 | 64.1 | 18.7 | 72.7 | 58.0–76.7 | 0.545[a] |
| | 1 | 14 | 66.9 | 19.4 | 72.8 | 52.8–86.4 | |
| | 1 | 14 | 66.9 | 19.4 | 72.8 | 52.8–86.4 | 0.006[b] |
| | ≥ 2 | 26 | 44.9 | 24.5 | 43.2 | 34.1–61.4 | |
| **KPS** | 0 | 20 | 84.2 | 12.6 | 88.5 | 81.3–92.0 | 0.500[a] |
| | 1 | 14 | 79.9 | 16.1 | 89 | 68.3–91.3 | |
| | 1 | 14 | 79.9 | 16.1 | 89 | 68.3–91.3 | 0.006[a] |
| | ≥ 2 | 26 | 63.7 | 17.8 | 63 | 48.8–78.8 | |
| **VISA-P** | 0 | 20 | 64.1 | 19.9 | 73 | 52.3–77.8 | 0.666[a] |
| | 1 | 14 | 65.5 | 22.0 | 74 | 47.8–83.3 | |
| | 1 | 14 | 65.5 | 22.0 | 74 | 47.8–83.3 | 0.003[a] |
| | ≥ 2 | 26 | 41.5 | 18.2 | 41.5 | 31.5–45.5 | |

Abbreviations: μ, media; SD, standard deviation; Med, median; IQR, interquartile range.

\* The *p* value indicates the statistical significance of the dependence between CP degree and the scale/item scores. The greater the difference in scores between those scales that correspond to knees with degree 0 and those that correspond to knees with degree 1, the greater the statistical significance. The same is true for degree 1 against degree ≥ 2.

[a] Mann-Whitney U test for non-paired samples.

[b] T-test for non-paired samples.

Questionnaires score ranges: 0–100.

**Table 9. Predictive capacity of questionnaires on degrees of chondromalacia patella.**

|  | Degree | Cut-off | AUC | CI95% | Sens (%) | CI95% | Spec (%) | CI95% | PPV (%) | CI95% | NPV (%) | CI95% |
|---|---|---|---|---|---|---|---|---|---|---|---|---|
| **KOOS-PF** | 1 | 53.4 | 0.751 | 0.592–0.910 | 78.6 | 52.4–92.4 | 73.1 | 53.9–86.3 | 61.1 | 38.6–79.7* | 86.4 | 66.7–95.3 |
| **KPS** | 1 | 71 | 0.761 | 0.602–0.920 | 78.6 | 52.4–92.4 | 69.2 | 50.0–83.5* | 57.9 | 36.3–76.9* | 85.7 | 65.4–95.0 |
| **VISA-P** | 1 | 48.5 | 0.784 | 0.625–0.943 | 78.6 | 52.4–92.4 | 80.8 | 62.1–91.5 | 68.8 | 44.4–85.8* | 87.5 | 69.0–95.7 |

Abbreviations: AUC, Area Under Curve; CI, Confidence Interval; Sens, Sensitivity; Spec, Specificity; PPV, Positive Predictive Value; NPV, Negative Predictive Value.

a Values ≤50%.

* The Degree column indicates the degree of CP that is analyzed. Therefore, the table indicates that those questionnaires/items that score higher than the cut-off indicate a higher probability of a degree 1 in CP.

Questionnaire score ranges: 0–100.

Note: all AUC CI95% values were >0.5.

The scores obtained in the VISA-P stood out from the other questionnaires as the one indicating the most severe stage of the pathology, in accordance with Visentini et al [38]. They stated that knee pathologies other than patellar tendinopathy, in particular patellofemoral pain, do not usually produce high VISA-P scores. This could be because a large part of the total score depends on the level of physical activity, yet many participants in this study were not physically active. Nevertheless, VISA-P showed a strong correlation with the other two PFP-specific questionnaires. Similarly, Hernández-Sánchez et al [49] found a strong correlation between VISA-P, in its adaptation to Spanish, and KPS, proving its validity and sensitivity for PFP. Overall, all questionnaires showed a strong correlation with each other ($r > 0.78$), with KOOS-PF values being the highest. The correlation of KOOS-PF with KPS showed values even higher than those obtained in other studies [30, 41]. Despite the good results, KOOS-PF showed significant differences between its two measurement moments.

In summary, KOOS-PF had the highest correlation, while VISA-P showed the least difference between the two measurement times. Consequently, the results do not confirm that one questionnaire is clearly better than another to assess PFP.

Regarding the content of the questionnaires, all three contained a similar number of questions. The vast majority focused on pain, a pathognomonic symptom of PFP and even

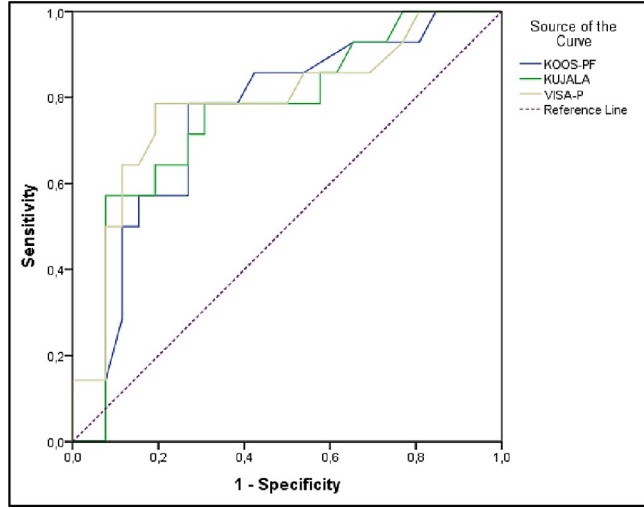

**Fig 2. ROC curves of the questionnaires according to the degrees of chondromalacia patella.** * Diagonal segments are generated by ties.

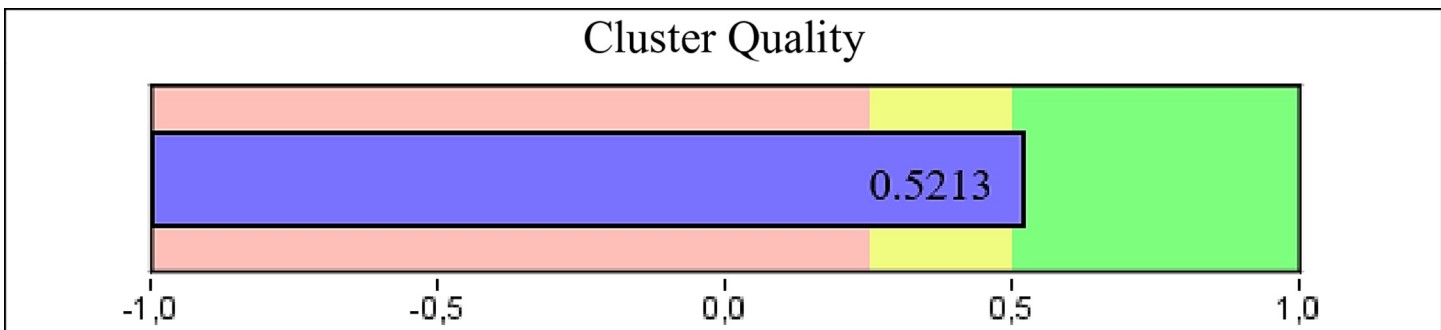

**Fig 3. Cluster quality as measured by the cohesion and separation silhouette.** Interpretation: red = poor, yellow = sufficient, green = good.

included in its definition [2, 4]. The item statements or response options usually contained the term "pain". However, in KPS-3, "*Walking. . .*", the term did not appear, raising doubts as to whether it referred to the ability to walk certain distances while in pain or without pain. Therefore, this study suggests always including the term "pain" in statements or answers in order to avoid confusion, and therefore bias.

Continuing with the "pain" component, the original KPS questionnaire included "*Slight pain when descending*" (stairs) but not when ascending them. Subsequently, the cross-cultural adaptation to Spanish [29] used in this study, introduced an additional possible answer in item 4,"*¿Podría subir y bajar escaleras*?" (translation: Could you ascend and descend stairs?*)*, giving the same score for pain when going up and down stairs. Consistent with Chinkulprasert et al [50], who demonstrated that eccentric contractions produce greater stress and patellofemoral distress than concentric contractions, several participants in this study stated that they tended to feel more pain descending than ascending. Consequently, we suggest that answers "*Dolor leve al subir escaleras*" (translation: Mild pain when ascending stairs) and "Dolor leve al bajar escaleras" (translation: Mild pain when descending stairs), should not be given the same score (8 points), since, if there is pain when ascending, i.e. with less stress on the joint, the pathology status would be more severe. Another option would be to separate the actions of ascending stairs and descending stairs into two items.

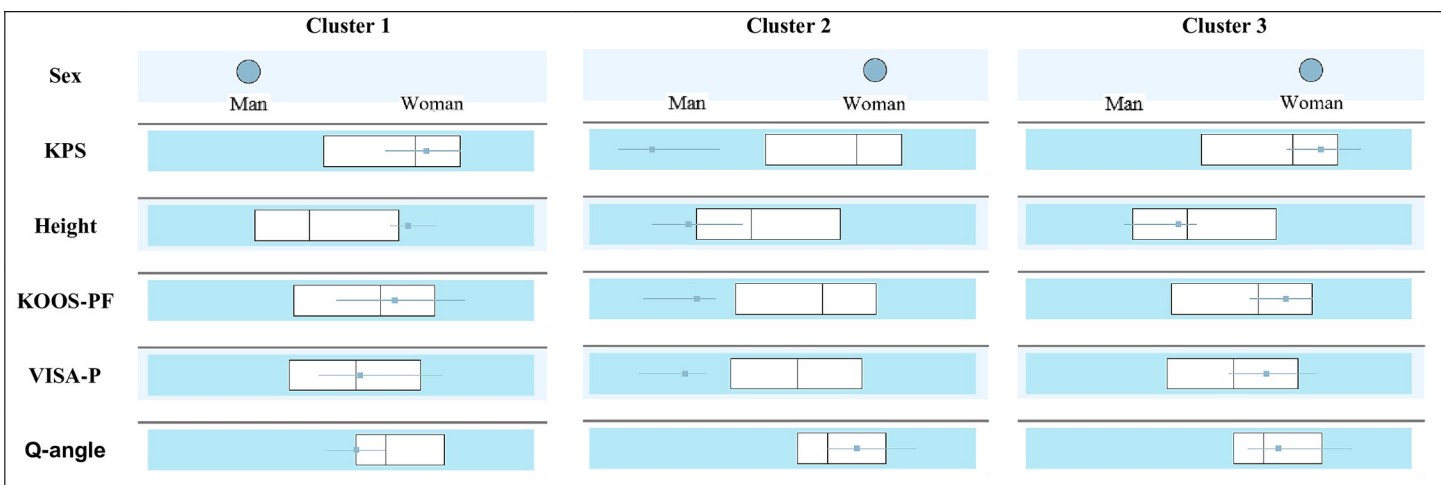

**Fig 4. Clusters derived from cluster analysis.** *The boxes indicate values for the whole sample (centre line = median, lateral ends = IQR) and the horizontal segments indicate values for the corresponding group (centre point = median, lateral ends = IQR).

The self-completion of KOOS-PF also gave rise to doubts. This questionnaire, as mentioned above, addresses the symptoms of the pathology during the last week. However, in items 2 (*"How often do you experience knee pain after stopping activity?"*) and 3 ("*How often does pain limit your activity?"*), the options "monthly" and "weekly" appear, confusing users. Perhaps it would be more appropriate to include: "Never", "Occasionally", "Some days", "Daily" and "Always".

Finally, the absence of questions about quality of life in PFP is striking, with the exception of KOOS-PF, which included the following question: "*Have you modified your sport or recreational activity due to your knee pain?*". Given the importance that several authors [5, 14, 16] give to the impact of PFP on quality of life, it seems logical to think that it should be considered by all questionnaires.

## About the physical assessment tests

In terms of inter-observer reliability, *patellar palpation test* and *squat test* had the best results. The former obtained two "good" results, while the squat was "excellent" in the first analysis and "good" in the second. Although the *squat* showed an "excellent" result, the variation between the two analyses did not allow it to be defined as the best test. This disparity is striking since the physiotherapist does not intervene. Even so, the *squat* is defined by many authors [2, 9, 31, 32] as the most relevant test for the assessment of PFP. Loudon et al [40] showed, like this study, that the test was not entirely reliable. This deficiency could be associated with the comments of several participants who stressed that one squat did not cause them pain, but that if they continued to do more, it would probably hurt. The fact that four squats were performed in the same session (although spaced apart in time), together with the other tests, may account for the variation in the measurement results, and hence the differences between the two inter-observer reliability analyses.

Another factor that needs to be taken into account in *squat test* performance is the bilaterality of the load exerted during the test. Subjects may unconsciously shift their weight towards the healthy limb in order to avoid pain, which does not occur in the case of bilateral involvement. The evaluator's control in this case is highly desirable. Consequently, the results would presumably show more positive results in the bilaterally affected group of participants compared to the unilaterally affected group. However, more positive results were obtained for unilateral patients, as was the case in the study by Loudon et al [40]. For all these reasons, these authors proposed an alternative unilateral test to the *squat test*, the step descent.

On the other hand, *squat* was the test with the most negative results and, consequently, appeared to be the least demanding.

The multiple antalgic compensations of the different parts of the body involved in the test are an influential factor, so the control of the correct execution is essential.

*Patellar tilt test* showed "acceptable" inter-observer reliability in both analyses. In this test, the physiotherapist has to detect a movement of a few degrees in a small structure such as the patella, i.e. it depends entirely on the perception and precision of the therapist. This aspect makes it difficult to agree on when it is positive or negative [51]. Although several authors [31, 32] consider the test to be unreliable, a finding confirmed in this study, it is frequently used at a clinical level due to its simplicity and rapid application.

*Clarke test* showed "good" inter-observer reliability in the first analysis, and "acceptable" in the second. In this test, it is difficult to establish a consensus on how much pressure to apply, which would justify the results obtained. In addition, Doberstein et al [39] highlighted the lack of clarity in the literature regarding its actual application mechanics, as well as the confusion

regarding when to consider it positive or negative. As with the *patellar tilt test*, some authors [2, 31] do not recommend its use.

In general, although the inter-observer reliability of the tests analyzed was at least "acceptable", clinical interpretation leads us to believe that this level of reliability is too low for a test to be applied in isolation or independently [32]. In the same vein, the authors of this study advocate greater precision in clinical assessments. In contrast, intra-observer reliability not only showed homogeneity between the two analyses performed, but they were all "excellent". Thus, they seem to be useful tools as long as the same therapist monitors the patient. It is difficult to get different therapists to agree on the pressure to be applied or degrees of mobility to be detected, and these tests are more useful when only one therapist applies them [51].

## About the relationships among functional assessment methods: Prediction capacity of questionnaires

To find scientific evidence on the relationship among functional assessment questionnaires [24, 27] or among physical tests [32] is not uncommon. However, this is not the case among questionnaires and physical tests, especially for PFP. This research complemented its results in a new way with statistics linking these two types of PFP assessment methods.

Thus, four of the physical tests analyzed, *patellar palpation*, *patellar tilt*, *Clarke* and *squat*, showed statistically significant relationships with the questionnaires. The relationship was logically with all questionnaires, given the strong correlation among them, i.e., subjects with high scores on the questionnaires would probably get negative tests. In fact, all values obtained were p<0.001 except for the *patellar tilt*, possibly due to their "acceptable" inter-observer reliability values.

Statistically significant results were also found regarding the specific items analysed associated with the *squat test*, the *patellar tilt test* and the *Clarke test*.

The item associated with patellar tilt, "*Flexion deficiency*" (KPS-13) was the least significant.

A positive result of this would imply a patella attached to the femur, and consequently stiffness, although clinically there are other factors such as the misalignment of the patella, its morphology or the inflammation of the knee itself which can also influence this stiffness [17]. As for the items associated with the *squat test* and the *Clarke test*, both stood out positively. On the one hand, *squat*-related items: "*Rising from sitting*" (KOOS-PF-4), "*Squatting*" (KOOS-PF-6), "*Stairs*" (KPS-4), "*Squatting*" (KPS-5), "*Do you have pain walking down stairs with a normal gait cycle?*" (VISA-P-2) and "*Do you have problems squatting?*" (VISA-P-5); clearly involved knee flexion-extension under load. On the other hand, the *Clarke*-related item, "*Do you have pain at the knee with full active non weight bearing knee extension?*" (VISA-P-3) generates an open kinetic chain contraction of the quadriceps to full extension causing the patella to bind to the femur, as well as aggressive rubbing [39].

In an innovative way, the results of this study showed cut-off points in the scores of the questionnaires that led to negative results when applying the physical tests.

Thus, the KPS cut-off points stood out as the highest (72.5–79.5) for predicting a negative test result. Likewise, it is striking that the questionnaire scores for *patellar palpation* and *patellar tilt* are always higher than for *Clarke* and *squat*. This could be because the latter two tests look for patellar involvement by gliding over the femoral groove (patellar tracking) with pain [39, 40], as opposed to *patellar tilt* and *patellar palpation*. That is, our results match *squat* with *Clarke*, as well as clearly indicating that a person with mild impairment is more likely to score positive on *Clarke* and *squat* than on *patellar tilt* and *patellar palpation*.

In relation to the predictive capacity of the questionnaires, *squat* was the most predictable physical test for both positive and negative results. Moreover, all three questionnaires were

good at predicting only negative results in the *Clarke test*; and KOOS-PF and KPS were good at predicting only positive results in *patellar palpation*. As the results on the relationships between assessment methods showed (see above), *patellar tilt* was the most difficult to predict. Finally, it should be noted that no single questionnaire was able to accurately predict the physical tests as a whole, i.e. the positive and negative results for *patellar palpation*, *patellar tilt*, *Clarke* and *squat*.

In relation to the specific items, none was a good predictor of the physical test with which it was associated. This is striking in the items associated with the *squat test*. KOOS-PF-6, which asks for "*Squatting*" pain, is only adequate to predict negative test results; and KOOS-PF-4, which asks for "*Rising from sitting*" pain, does not correctly predict any results. The difference between them can be explained by the fact that KOOS-PF-4 assesses only concentric extension of the knee after a period of inactivity and knee flexion. In fact, these two items have a weak correlation between them, which justifies them both being in the same questionnaire. With regard to KOOS-PF-6, the results are especially striking as it asks for the *squat test* gesture. However, some subjects commented, regarding squats, that one squat would not hurt, but doing several in a row would, while others said that the pain decreased or disappeared as they did more. In KOOS-PF-4, the lack of good results could be due to the fact that only a concentric contraction of the quadriceps is performed in the eccentric phase when the amount of friction of the patella with the femur doubles and the cartilage suffers more pressure [52]. KPS-5, which asks about the functional limitation caused by squatting pain, and VISA-P-5 ("*Do you have problems squatting*?"), which addresses the intensity of pain in the same gesture, are excellent predictors of negative *squat test* results. It is possibly not a good predictor of positive results because, when translating both items into English, the term "*squat*" was replaced by "*cuclillas*", which is associated with a deep squat (maximum knee flexion), instead of "*sentadilla*", which implies only 90˚ flexion [40]; moreover, when squatting, the load is on the forefoot, increasing patellofemoral pressure more than in a squat. Therefore, they would not be optimal items for predicting a positive *squat test* result. Finally, two other items related to stairs were associated with *squat test*: pain ascending or descending "*Stairs*" (KPS-4); and "*Do you have pain walking downstairs with a normal gait cycle*?", asking for pain intensity (VISA-P-2). Although both refer to stairs, they differ in that one considers the ability to perform the task and the other the pain of performing the task. This is in addition to the aforementioned greater patellofemoral involvement when descending stairs compared to ascending stairs. Thus, an inverse correlation was found between the two items. Regarding predictive ability, both KPS-4 and VISA-P-2 were shown to be good predictors of good predictors for the *squat test*, but only for negative results, since squatting is a less aggressive activity for the patellofemoral joint than stairs. i.e., squatting carries bilateral loading within the base of support while ascending or descending stairs implies unilateral loading outside the base of support.

As for VISA-P-3, "*Do you have pain at the knee with full active non weight bearing knee extension*?", its association with the *Clarke test* was analysed as it is a gesture of quadriceps contraction in open kinetic chain up to full extension. However, it is not a good predictor of a positive *Clarke test*. We consider the *Clarke test* to be a significantly more severe test than the knee extension without pressure from the assessor, which is also very subjective. The inter-observer reliability median results of this study supported this idea.

## Predictive capacity of questionnaires on MRI-diagnosed CP grades

Although this study assesses subjects with PFP in general, a representative subsample was diagnosed with CP by MRI. Thus, the relationships obtained between degrees of CP were assessed regarding the scores on the three questionnaires. No statistically or clinically significant differences were found between grades 0 and 1, in agreement with Thomas et al [22] and Flanigan

et al [53], who considered MRI to be an inaccurate tool for the diagnosis of CP, especially for low grades [22]. However, we did find differences between grades 1 and above, i.e. 2 and even 3 and 4, with severe structural damage. Furthermore, there is evidence of low correlation between grades and symptoms [22], the latter associated with functional assessment questionnaires. Other recent evidence, 2020 [12], with a substantial sample size (n = 230), even found asymptomatic subjects with structural damage of the patellofemoral cartilage on MRI in 57% of cases. This would justify the poor relationship between the questionnaires and the MRIs. Consequently, the predictive ability of the questionnaires on CP grades set by this objective method was not particularly good. According to the previous paragraph, it was not possible to determine the relative predictive ability for the pair grades 0 and 1. For the pair 1 and greater than 1, only KOOS-PF and VISA-P had a relative predictive capacity. According to the cut-off points, scores below 53.4 and 48.5, respectively, predicted grades higher than 1. However, the 95% IC of the PPV was not adequate, so the predictive capacity for grade 1 should be considered with caution until a larger sample size is available.

As with the results discussed in previous sections, this analysis yields new data that lead therapists to decide whether or not to apply certain methods, but also to encourage appropriate combinations among them, in this case among questionnaires and tests that, because of their relationship, confirm an assessment or diagnosis.

This study advocates the efficiency of functional assessment questionnaires, although it understands that they are insufficient in this case as they are patient reported outcome measures (PROMs). That is, they provide information on symptomatology and only consider the subjectivity of the user. Similarly, specific physical tests do not consider the multifactorial aspect of PFP, as mentioned in the introduction, focusing primarily on the presence of pain. We believe it is essential to collect information on the factors that influence the development of the pathology. Thus, a holistic and individualized assessment would allow the establishment of medical (i.e. pharmacological, surgical), orthopaedic (insoles, knee braces) and physiotherapeutic (i.e. electrotherapy, massage therapy, therapeutic taping, therapeutic exercise) goals and procedures, associated both to the symptomatology and to the possible existing structural damage, but also to the factors that work against PFP. For example, if there is patellar hypermobility due to quadriceps weakness, this muscle should be strengthened. Or if there is patellar lateralisation caused by asymmetry between the muscle tone of the vastus externus and quadriceps internus or retraction of either of them, they should be balanced to re-centre the patella. In this way, the functional recovery of the user would be optimized and aggravations, relapses and sequelae could be prevented.

## Association among functional assessment methods, Q-angle, gender and height

Although the cluster analysis included all methods of functional assessment, the presence of physical tests prevented the clusters from being appropriate, and therefore logical. Therefore, the statistical test ruled them out. This fact, together with the associations obtained between the questionnaires, gender, height and Q-angle, led us to believe that the questionnaires assessed are more reliable methods to be applied in PFP and/or CP.

Thus, the results clearly indicated a differentiation by gender and height. Both variables, according to authors such as Kasitinon et al [54], influence the Q-angle, and this in turn, the patellofemoral involvement. Men are associated with a lower Q-angle than women and tall people are associated with a smaller Q-angle than short people [54]. As for the three clusters generated, two of them (clusters 1 and 3) included subjects with healthier outcomes according

to the scores of all questionnaires. One of the groups obtained consisted of relatively tall men and therefore with a reduced Q-angle (cluster 1).

The other two groups were made up of women, with the taller women (cluster 3) having a lower Q-angle than the shorter women (cluster 2). However, the group of men (taller than tall women) had a smaller Q-angle than tall women.

These findings would also enable practitioners to predict the trend of questionnaire scores, i.e. intensity of symptomatology, from simple anthropometric data linked to gender. Consequently, this information would make it easier to prevent an unfavourable evolution of the pathology, especially in early stages.

Regarding the strengths of the study, questionnaires that, although validated, did not have cross-cultural adaptations in Spanish were excluded, while maintaining an appropriate level of methodological quality. In fact, this led to a limitation, namely the exclusion of validated and specific questionnaires in common use.

Prospectively, other cross-cultural adaptations of questionnaires of scientific interest should be carried out, both into Spanish and other languages. Furthermore, this study suggests the creation of assessment protocols that take into account factors that negatively influence PFP, enabling them to be minimized during the functional recovery process. Finally, the results obtained in the analysis regarding MRI-mediated CP grades should be treated with caution due to the size of the subsample (see above). On the basis of this new limitation, we propose the consideration of a large sample that provides robust evidence.

## Conclusions

Regarding the questionnaires, KOOS-PF, KPS and VISA-P demonstrated their diagnostic value in "patellofemoral pain" and/or "chondromalacia patella", based on their validity, reliability, sensitivity, specificity, predictive value and clinical applicability.

All of them showed their predictive capacity and logical groupings by clusters, with respect to gender, height and Q-angle. Taller women were associated with a higher Q-angle and higher scores on all questionnaires.

No single questionnaire could accurately predict all physical tests in their entirety, i.e., the positive and negative results of *patellar palpation*, *patellar tilt*, *Clarke* and *squat*.

KOOS-PF and KPS were better overall predictors of the physical tests, although KOOS-PF was more demanding due to its lower cut-off points, and therefore more suitable for mild symptomatology and functional limitations. All three were suitable predictors of negative *squat* and *Clarke tests*. They were also good predictors of chondromalacia patella grades by magnetic resonance imaging, with KOOS-PF and VISA P being better predictors of grades greater than one, with intermediate cut-off points.

Although the questionnaires analysed were found to be clinically feasible, KOOS-PF appeared to be the most versatile of the three, and, in general, the most appropriate in mild cases and for early detection and prevention. KPS is the most advisable in severe cases.

With regard to the physical tests, the *squat test* was the most appropriate and stood out positively for its reliability and its clinical relationship with the questionnaires, which predicted it correctly. However, of the items associated with it, not even the one that asks directly about pain with such a gesture, KOOS-PF-6, offered good predictive capacity for negative test results. For this reason, and because the other physical tests are not fully predicted by the questionnaires, we suggest that all physical tests be complemented by at least one assessment questionnaire, choosing the most appropriate according to the context.

Furthermore, with the exception of the *squat test*, it should preferably be applied by a single examiner according to the levels of inter-observer reliability found, or at least be complemented by other methods.

This study also advocates improving, clarifying and unifying the definitions and interpretations of the physical tests, especially the *squat test*, and the items associated with this gesture, in order to achieve results with greater scientific rigour.

Height, Q-angle and gender are simple data that can predict and prevent the unfavourable evolution of the pathology, especially in the early stages.

In general, functional assessment methods, although specific to PFP, should be applied by combining them with each other. They should also be complemented by data on PFP influencing factors. Moreover, imaging tests could be useful if structural damage to the cartilage is suspected.

## Supporting information

**S1 Data. Matrix including outcomes about study variables.**
(XLSX)

## Acknowledgments

The authors would like to thank the Research Group "Area of Physiotherapy CTS-305" of the University of Seville, Spain; for its contribution in this study.

## Author Contributions

**Conceptualization:** Gema Chamorro-Moriana, Carmen Ridao-Fernández.

**Data curation:** Fernando Espuny-Ruiz, Carmen Ridao-Fernández.

**Formal analysis:** Fernando Espuny-Ruiz, Eleonora Magni.

**Investigation:** Gema Chamorro-Moriana, Eleonora Magni.

**Methodology:** Gema Chamorro-Moriana, Eleonora Magni.

**Project administration:** Fernando Espuny-Ruiz, Carmen Ridao-Fernández.

**Resources:** Gema Chamorro-Moriana, Carmen Ridao-Fernández.

**Supervision:** Gema Chamorro-Moriana.

**Validation:** Carmen Ridao-Fernández, Eleonora Magni.

**Visualization:** Carmen Ridao-Fernández.

**Writing – original draft:** Gema Chamorro-Moriana, Fernando Espuny-Ruiz, Eleonora Magni.

**Writing – review & editing:** Fernando Espuny-Ruiz, Carmen Ridao-Fernández, Eleonora Magni.

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
