## [Decision Letter · Decision Letter 0]

19 Feb 2024

PONE-D-23-41403Clinical value and appropriateness of functional assessment methods for patellofemoral pain: Reliability and predictive and preventive capacity of questionnaires & physical testsPLOS ONE

Dear Dr. Espuny-Ruiz,

Thank you for submitting your manuscript to PLOS ONE. After careful consideration, we feel that it has merit but does not fully meet PLOS ONE’s publication criteria as it currently stands. Therefore, we invite you to submit a revised version of the manuscript that addresses the points raised during the review process.

We look forward to receiving your revised manuscript.

Kind regards,

Mehrnaz Kajbafvala, Ph.D

Academic Editor

PLOS ONE

Journal Requirements:

Reviewers' comments:

Reviewer's Responses to Questions

**Comments to the Author**

1. Is the manuscript technically sound, and do the data support the conclusions?

Reviewer #1: Yes

Reviewer #2: Yes

2. Has the statistical analysis been performed appropriately and rigorously? 

Reviewer #1: Yes

Reviewer #2: Yes

3. Have the authors made all data underlying the findings in their manuscript fully available?

Reviewer #1: Yes

Reviewer #2: Yes

4. Is the manuscript presented in an intelligible fashion and written in standard English?

Reviewer #1: Yes

Reviewer #2: Yes

5. Review Comments to the Author

Reviewer #1: This study was aimed to determine the most appropriate method of functional assessment for "patellofemoral pain" (PFP)/“chondromalacia patella” for its diagnostic value, (validity, reliability, sensitivity, specificity, predictive value and clinical applicability); to outline initial interpretations of the questionnaires and their appropriateness, through the cutoff points determined in their scores based on physical test and Magnetic Resonance Imaging (MRI); to establish which methods should be used in conjunction with each other to obtain clinical diagnoses that are robust effective and efficient. The study findings indicated that KOOS-PF, KPS and VISA-P demonstrated their diagnostic value in PFP/chondromalacia (validity, reliability, sensitivity, specificity, predictive value and clinical applicability). KOOS-PF was the most versatile, and the most appropriate in mild cases and for early detection and prevention. Squat was the best due to its reliability and clinical relationship with the questionnaires, which predicted it correctly. The methods discussed should be applied in combination. Overall, the study is interesting, however there are several clarifications needed.

Comment#1

Title is not appropriate. Please state title more accurately, scientifically and concisely.

Comment#2

Please edit Keywords. Keywords should be the reflective of the research performed. In addition to predictive capacity, intra- and inter-observer reliability and clinical usefulness were assessed in this study.

Comment#3

Introduction. Please insert reference/references in different parts of Introduction section for supporting the sentences stated.

Comment#4

Methods. How did you assess the preventive capacity of questionnaires & physical tests?

Comment#5

Please state sample size calculation.

Comment#6

Please state study limitations.

Reviewer #2: Thank you for providing us with the opportunity to review this manuscript. Overall, the topic is a nice job with some minor concerns, which I will outline below.

Line 52. What do you mean?

Line 83-85. Please rewrite these sentences.

Line 93. “so non-specific questionnaires, such as the Fulkerson Knee Instability Scale (FKIS)[26], are also.. “What do you mean?

Line 129. Please clearly explain PFP diagnosis in your study participants.

Did having musculoskeletal disorders in other joints as an exclusion criteria? Please explain exclusion criteria in detail.

Line 143. “KPS” is “KOOS-PF”?

Line 145-146. Please edit this sentence grammatically.

Line 168. Please explain words before abbreviating “FE-CR, FE-CR)

Please explain CP grad using MRI assessment.

6. PLOS authors have the option to publish the peer review history of their article (what does this mean?). If published, this will include your full peer review and any attached files.

Reviewer #1: No

Reviewer #2: No

---

## [Author Response · Author response to Decision Letter 0]

9 Mar 2024

Dear Mehrnaz Kajbafvala, Ph.D, Academic Editor,

We appreciate your response to our manuscript and the effort you have made to improve it. We have carefully read all your comments and those of the reviewers, responding to each of them point by point.

Thanks to your expert suggestions we have revised the manuscript, clarified some aspects of the text, made grammatical modifications, corrected some mistakes, attached the required documents (database as a supplementary file and informed consent; informed assent of the minor and ethics committee approval for the journal), etc. All modifications to the Main Text have been made with track changes.

We believe that the current manuscript is more complete and rigorous than the original. We hope we have met the expectations of your prestigious journal.

Our responses are shown point by point below. We remain at your disposal for any further suggestions.

Best regards,

Fernando Espuny-Ruiz, MSc (corresponding author 1)

Gema Chamorro-Moriana, PhD (corresponding author 2)

---

## [Decision Letter · Decision Letter 1]

1 Apr 2024

Clinical value of questionnaires & physical tests for patellofemoral pain: validity, reliability and predictive capacity.

PONE-D-23-41403R1

Dear Dr. Fernando Espuny-Ruiz,

We’re pleased to inform you that your manuscript has been judged scientifically suitable for publication and will be formally accepted for publication once it meets all outstanding technical requirements.

Kind regards,

Mehrnaz Kajbafvala, Ph.D

Academic Editor

PLOS ONE

Additional Editor Comments (optional):

Reviewers' comments:

Reviewer's Responses to Questions

**Comments to the Author**

1. If the authors have adequately addressed your comments raised in a previous round of review and you feel that this manuscript is now acceptable for publication, you may indicate that here to bypass the “Comments to the Author” section, enter your conflict of interest statement in the “Confidential to Editor” section, and submit your "Accept" recommendation.

Reviewer #1: All comments have been addressed

Reviewer #2: All comments have been addressed

2. Is the manuscript technically sound, and do the data support the conclusions?

Reviewer #1: Yes

Reviewer #2: Yes

3. Has the statistical analysis been performed appropriately and rigorously? 

Reviewer #1: Yes

Reviewer #2: Yes

4. Have the authors made all data underlying the findings in their manuscript fully available?

Reviewer #1: Yes

Reviewer #2: Yes

5. Is the manuscript presented in an intelligible fashion and written in standard English?

Reviewer #1: Yes

Reviewer #2: Yes

6. Review Comments to the Author

Reviewer #1: (No Response)

Reviewer #2: (No Response)

7. PLOS authors have the option to publish the peer review history of their article (what does this mean?). If published, this will include your full peer review and any attached files.

Reviewer #1: No

Reviewer #2: No

---

## [Editor Report · Acceptance letter]

4 Apr 2024

PONE-D-23-41403R1 

PLOS ONE

Dear Dr. Espuny-Ruiz, 

I'm pleased to inform you that your manuscript has been deemed suitable for publication in PLOS ONE. Congratulations! Your manuscript is now being handed over to our production team.

Kind regards, 

on behalf of

Dr. Mehrnaz Kajbafvala 

Academic Editor

PLOS ONE